# Effects of an Isobutylene–Maleic Anhydride Copolymer on the Rheological Behavior and Early Hydration of Natural Hydraulic Lime

**DOI:** 10.3390/polym14194104

**Published:** 2022-09-30

**Authors:** Xiaohong Wei, Yunfeng Li, Jing Hui, Wenwen Wang, Biao Zhang, Liangliang Chang, Yuhong Jiao, Zhen Sang, Hongjie Luo, Xiufeng Wang

**Affiliations:** 1Shaanxi Key Laboratory of Green Preparation and Functionalization for Inorganic Materials, Institute of Silicate Cultural Heritage, School of Materials Science and Engineering, Shaanxi University of Science and Technology, Xi’an 710021, China; 2Research Institute of Cultural Relics, Shanghai University, Shanghai 200444, China

**Keywords:** natural hydraulic lime, isobutylene–maleic anhydride copolymer, rheological behavior, dispersion capacity, adsorption, early hydration

## Abstract

Natural hydraulic lime (NHL) is a cementitious material widely used in the restoration of stone cultural relics and maintenance of historic buildings, the practical use of which is mainly hindered by its poor fluidity. Due to the multilayer (double-layer) adsorption that isobutylene–maleic anhydride (IBMA) has on the surface of NHL, the effects that IBMA copolymer have on the fluidity and hydration of NHL were thus investigated. Moreover, the yield stress and plastic viscosity of NHL pastes were found to be reduced significantly by the incorporation of IBMA. Combined with the effects of electrostatic repulsion and steric hindrance, the flocculated structures in NHL pastes were gradually dismantled, releasing the trapped water and leading to a significant enhancement in the fluidity of NHL. IBMA was found to postpone the early hydration of NHL. In particular, it showed that adding specific content of IBMA can significantly improve the early strength of NHL.

## 1. Introduction

Cracks, a form of serious damage observed in stone cultural relics, have received wide research attention. These cracks in the structures of stone cultural relics are grouted to achieve their renovation and improve their stability. However, the materials used in grouting need to exhibit excellent initial fluidity and permeability, in addition to good injection and consolidation strength [1]. Therefore, the development of grouting materials plays an essential function in repairing cracks in stone cultural relics by grouting.

The history of NHL materials as building gelling materials dates back to ancient times [2,3]. In recent years, natural hydraulic lime (NHL) is the most promising alternative material to cement and epoxy resin for stone cultural relics’ restoration and has drawn significant research and cultural heritage conservation attention. At present, NHL-based pastes are widely used in the restoration of cultural relics due to the good properties of NHL, such as low shrinkage, low soluble salt content, moderate strength, and good compatibility with the matrix of the cultural relics that require restoration [4,5,6]. Moreover, when hardening in air, NHL2NHL pastes carbonate by absorbing carbon dioxide (CO_2_) from the air, leading to a reduction in CO_2_ in the environment, indicating that NHL could be used in environmental remediation [7]. In view of its properties, NHL is thus expected to be of wide use in the restoration of historical artifacts. However, the poor fluidity and low early strength of NHL-based pastes hinder their practical use in the grouting and restoration of stone cultural relics. Therefore, it is an urgent challenge to improve the properties of NHL paste in fluidity, permeability, and early strength so that it can be pourable enough to repair cracks in stone cultural relics. Since the grouting and solidification process of NHL pastes are similar to those of concrete and mortar, the issues associated with NHL pastes can be solved using similar solutions to remedy issues with concrete and mortar. As a result, any variation in the properties of NHL can be achieved by mixing it with reinforcement materials and chemical admixtures, such as superplasticizers, to adjust and control its fluidity, permeability, and strength [8,9,10,11]. Currently, issues with the (early) strength of concrete and mortar are mainly addressed by adding reinforcement materials such as silica powder and fiber, fluidity and permeability by adding admixtures such as water-reducing agents. Therefore, to improve the early strength of NHL, a variety of reinforcement materials such as silica fume [12], metakaolin [13,14], fiber [15,16,17,18] and graphene oxide (GO) [19] have been used to improve its mechanical properties and durability. Luis G Baltazar et al. [20] found that the fluidity, permeability, and strength of NHL paste could be improved by the addition of varying amounts of a high-efficiency water-reducing agent and micro-nano silica powder. Fernando Jorne et al. [21] analyzed the effects that the addition of a superplasticizer, resting time, and grouting pressure had on the grouting properties of NHL, and constructed equipment to simulate grouting. In spite of extensive research work having been undertaken, the fluidity and early strength issues of NHL have yet to be fully resolved, which has limited the application of NHL in the restoration of stone artifacts. Therefore, it is still important to resolve the issues associated with the properties of NHL.

Isobutylene–maleic anhydride (IBMA) copolymer is a polymer with alternating isobutylene and maleic anhydride units that is soluble in alkaline solution. Research has shown that IBMA copolymer exhibits good dispersibility and gelling effect on inorganic micro–nano particles, and that it is also a promising superplasticizer for NHL paste. Moreover, IBMA has also been widely used as a ceramic gel casting agent. Recently, Shimai et al. [22,23] found that when IBMA is added to aqueous aluminum oxide (Al_2_O_3_) suspension at room temperature, it exhibits dispersion and gelation properties, phenomena that have also been observed in other ceramic powder systems [24,25,26]. The results thus show that IBMA copolymer can be adsorbed on the surface of inorganic particles, generating electrostatic repulsion and steric hindrance that promote the dispersion of its flocculated or agglomerated structure, thus improving the dispersion of the system. However, the mechanism of the particle dispersion of NHL and its gelation enhancement by the IBMA copolymer have not yet been studied.

Therefore, in this study, the effect that IBMA copolymer has on the rheological properties, dispersion, hydration process, and early strength of NHL, in addition to the underlying mechanism, was studied. The influence of IBMA copolymer on the rheological properties of NHL paste was investigated by fluidity and stress-viscosity tests and the results were analyzed. By observing and analyzing the adsorption behavior, the zeta potential, and microstructure of IBMA on the surface of NHL particles in paste samples, the dispersion mechanism of IBMA on the NHL particles was revealed. Moreover, the effect and enhancement of the early hydration of NHL by IBMA were studied.

## 2. Experimental

### 2.1. Materials

NHL2 was procured from Hessler Kalkwerke GmbH (compliant with European standard EN-459-1) and its chemical composition was analyzed by X-ray fluorescence spectrometry, as shown in Table 1. IBMA (IBMA-104, molecular weight: 55,000–60,000) was purchased from Kuraray Co., Ltd. (Tokyo, Japan), with the molecular structure shown in Figure 1.

### 2.2. Sample Preparation

A UJZ-15 solid admixtures mixer was used to prepare paste samples. Before mixing, a solution of IBMA in deionized water was prepared. After adding the IBMA solution to NHL, the paste was mixed at low speed for 2 min and then at high speed for 3 min to achieve a consistent mixture, wherein the mass ratio of water to NHL in the prepared NHL samples was fixed at 1:2. After mixing, the fresh NHL paste was immediately poured into a standard 40 × 40 × 40 mm^3^ mold and demolded after 24 h of curing. The demolded samples were then kept in a temperature humidity chamber with 60% relative humidity at 20 °C for 7 days and 28 days respectively, to investigate the hydration properties and performance analysis of the hardened pastes.

### 2.3. Testing Methods

#### 2.3.1. Adsorption Amount

The changes in the carbon concentration in the IBMA solution before and after its addition to NHL were assessed to determine the IBMA adsorption isotherm under equilibrium conditions. To achieve this, paste was centrifuged at 4500 rpm for 10 min to extract the interstitial fluid, which was then filtered through a 0.45-μm nylon filter. The filtered interstitial fluid was diluted with deionized water to detect the total organic carbon (TOC) using a TOC-VCPH total organic carbon analyzer (Shimadzu, TOC-VCPH, Tokyo, Japan).

#### 2.3.2. Zeta Potential

The zeta potential of the particles in the paste samples was determined using a ZEN3690 zeta sizer (Malvern Instruments, Malvern, UK). Diluted paste (0.1 g of paste dissolved in 50 mL of water) was shaken at 25 °C for 5 h to achieve adsorption equilibrium, whereafter the solution was allowed to stand for 1 h. The supernatant liquid obtained from the diluted paste comprising stable NHL particles and water was then used in the zeta potential measurements. For each sample, 10 measurements were made and the average zeta potential value was used in this study.

#### 2.3.3. Fluidity

To investigate the flow retention capacities of paste samples with different IBMA content, the fluidity of the pastes over time was measured. The dispersion performance of IBMA was determined to identify the fluidity of paste. According to the Chinese national standard GB/T 8077-2000 (test method for the homogeneity of concrete admixtures), the fluidity of the paste samples (W/C = 0.5) was measured by micro-slump conemethod. During the test, the fresh NHL paste with IBMA copolymer was filled into a micro slump cone with a bottom diameter of 60 mm, and then lifted the cone quickly. While the paste had fully spread, recording the two vertical cross diameters of the spread paste to indicate its fluidity property.

#### 2.3.4. Rheological Properties

The rheological properties, including yield stress and plastic viscosity, of the NHL paste were evaluated using a HAAKE^TM^ MARS^TM^ rotary rheometer. The test mold is a parallel-plate geometry with a rough surface of 35 mm diameter. 2 mm gap between the plates was used for the test. The test protocol consisted of pre-shearing for 60 s at 1 s^−1^, then followed by a resting step of 60 s. This step was conducted to ensure that all of the samples were in a similar initial state. After resting, the shear rate was then increased from 0 s^−1^ to 200 s^−1^ in 200 s, and the flow curve was recorded. All pastes were tested at 20 °C. The yield stress and plastic viscosity of fresh paste samples was studied by Bingham (Equation (1)) model.
(1)$${\tau =\tau }_{f\;}{+\eta }_{p}\times \;\gamma$$
where *τ* is the shear stress (Pa), $\tau_{f\;}$ is the yield stress (Pa), $\eta_{P}\;$is the plastic viscosity (Pa.s), and *γ* is the shear rate (s^−1^).

#### 2.3.5. Hydration of NHL Pastes

The heat emission of fresh paste samples was monitored to study the effect that the IBMA copolymer had on the hydration of NHL. A TAM Air microcalorimeter was used to monitor the exothermic rate of the hydration of NHL in real time with an experimental accuracy of 20 μW. Prior to conducting any measurements, the instrument was calibrated and kept at 20 °C for 24 h. In the experiments, on-line water addition and in-situ stirring were used, with the NHL and IBMA solutions mixed together in ampoules to prepare paste samples, where the amounts of copolymer in the paste samples were 0 wt%, 0.3 wt%, 0.5 wt%, and 0.8 wt%. The heat release properties of the samples were measured over 168 h.

To further evaluate the hydration process and determine the final nature of the hydrate, different test methods were used. The phase compositions of the composites with different organic content was qualitatively analyzed by X-ray diffractometer (XRD, Bruker D8 advance, Karlsruhe, Germany) using Cu Kα radiation (λ = 1.5406 Å), operated under 40 kV and 40 mA over a 2θ scan range of 10–75° with a step size of 0.02°. The infrared absorption spectra of the samples over a range of 400–4000 cm^−^^1^ were determined by a Fourier-transform infrared spectrometer (FT-IR, Bruker Vector-22, Karlsruhe, Germany) to analyze the changes in calcium hydroxide, calcium silicate hydrate, and other hydration products of the pastes. The variation in the chemical compositions and electronic structures of the paste samples before and after adsorption was investigated by X-ray photoelectron spectroscopy (XPS, Shimadzu, AXIS SUPRA+, Japan).

#### 2.3.6. Microstructure Analysis

The microscopic morphologies of NHL particles in water and IBMA solutions were analyzed using a VHX-7000 digital microscope (Keyence). For the aqueous NHL system, after mixing the NHL and water in a mass ratio of 1:200, 2 mL of the middle layer of the paste were dropped on a glass slide to observe its morphology. For the IBMA solution system, according to water: NHL mixing ratio of 1:2, IBMA/NHL = 0.2%, 0.4%, and the mass ratio 1:200 of IBMA to NHL solution, low and high concentration IBMA solutions were prepared. After mixing to achieve uniformity of the samples, 2 mL of the middle layer paste were dropped on a glass slide to observe their morphology.

#### 2.3.7. Mechanical Properties

The mechanical properties of the samples were tested using a 1036 PC universal testing machine (Taiwan Baoding Instrument Co., Ltd., Taiwan), with a maximum test load of 10 kN and a loading rate of 0.01 kN s^−1^). To study the influence IBMA has on NHL, the compressive strength of the pastes samples was measured after 7 and 28 days’ curing period.

## 3. Results and Discussion

### 3.1. Adsorption of IBMA on the Surface of NHL

To explore the mechanism unpinning the influence that IBMA had on the fluidity of a paste of NHL, the adsorption behavior of IBMA on the surface of NHL particles was studied. Figure 2 shows the adsorption curve of IBMA on the surface of NHL particles, from which it can be seen that there are two adsorption saturation plateaus in the adsorption data of IBMA. Therefore, IBMA is adsorbed in multiple layers on the surface of the NHL particles, potency similar to PCE on the surface of cement particles [27]. It is evident from the adsorption isotherm of IBMA shown in Figure 2 that the first plateau is reached at low concentration, with the second plateau observed after further adsorption at higher concentration. In the adsorption curve, the IBMA content in the sample that exhibited adsorption saturation is referred to as the adsorption saturation amount. It is also evident from Figure 2 that the adsorption saturation amount increased from 12 mg/g to 26 mg/g from the first layer to the second layer, due to the content of IBMA increasing from 0.6 wt% to 1.3 wt%. As observed, the adsorption capacity increases rapidly in line with an increase in the IBMA content of the sample. When vacancies are still present on the surface of the NHL particles the adsorption process continues. However, after saturation is reached, the adsorption capacity no longer changes with an increase in the concentration of the IBMA copolymer. Based on this analysis, such an adsorption curve shows that IBMA is attached to NHL particles by multilayer surface adsorption, similar to the adsorption of IBMA on cement particles [27]. However, to date, no studies have elucidated the multilayer adsorption mechanism of IBMA on the surface of NHL particles.

The adsorption of IBMA on NHL particles has a significant effect on the zeta potential of the NHL pastes depending on the adsorption capacity of the chemical substance being tested. The zeta potential measurement of a NHL paste containing IBMA is shown in Figure 3, which shows a positive zeta potential value approximately +5 mV. The reason for a positive potential being observed is the abundance of Ca^2+^ ions in NHL suspension leads to their adsorption on NHL particles. However, upon the addition of IBMA, the zeta potential drops significantly from a positive to a negative value, and then the potential stops falling as adsorption saturation is reached, resulting in a zeta potential value of close to −15 mV. The negative value of NHL paste can thus be attributed to the adsorption of negatively-charged IBMA on the NHL particles via electrostatic attraction. With the further addition of IBMA, although the adsorption continues to increase, the potential value tends toward being stable. This indicates that the second layer of IBMA adsorbed on the NHL particles does not lead to any further changes in the zeta potential of the NHL paste, which resembled the previously reported studies [28,29,30]. As mentioned above, the high adsorption capacity of the polymer (IBMA) gives rise to a high negative zeta potential, which leads to a higher electrostatic barrier effect and contributes to the higher dispersion ability.

According to Mollah’s charge-controlled reaction model, multilayer adsorption may occur via the following process. As schematically shown in Figure 4, IBMA reacts with hydroxyl groups in the alkaline environment to generate carboxyl groups via the ring opening of acid anhydride groups. At the same time, ammonium groups enter the solution and are exposed to carboxylic acid groups. The abundance of –COO^–^ groups in the chains of IBMA makes it negatively charged under alkaline conditions. In hydrated NHL pastes, the surface of positively-charged minerals such as C_2_S, C_3_A, etc. can attract IBMA to it by electrostatic attraction. As the low content IBMA amount increases, the adsorption on the first layer of IBMA to the surface of NHL particles quickly completes, resulting in the adsorption saturation of the first plateau. At the same time, the existing of a large amount of Ca^2+^ ions in the intricate pore solution with the –COO^–^ groups of the polymer chain forms a relatively stable Ca^2+^ layer on the first layer of adsorbed IBMA, which in turn triggers the adsorption of the second layer of IBMA. As shown in Figure 4, at high content, IBMA is saturated again and enters the second plateau. It can thus be concluded that the first adsorption isotherm plateau indicates the first layer adsorption of IBMA completely covers the NHL surface. The second plateau thus represents the saturation adsorption layer.

### 3.2. Fluidity Measurements

The fluidity measurements of NHL pastes with different IBMA content ranging from 0.1 wt% to 2.0 wt% (solid content to NHL) are shown in Figure 5 and Figure 6. In Figure 5, the increasing IBMA addition can be seen to increase the diameter of the spread of the NHL pastes from 60 mm to a maximum value of 220 mm (at 0.6 wt% IBMA addition). It can thus be concluded that IBMA plays an important role in improving the fluidity of the NHL pastes. The dramatic increase in the fluidity of the NHL pastes can be attributed to the decomposition of the flocculated NHL structure and the subsequent release of its encapsulated water. After this process occurs, the spread diameter is basically unchanged upon the further addition of IBMA, consistent with the action of that a superplasticizer has been observed to have on fcps [27]. This activity indicates that a further increase in the IBMA content has no further effect on the fluidity of NHL, which remains stable upon the further addition of IBMA.

Moreover, Figure 6 shows the impact that IBMA has on the fluidity of NHL pastes over time. It can be observed that the diameter of the spread of NHL pastes with different IBMA content decreases with an extension in the time. However, with an increase in the IBMA content, the changes in the spreading diameter gradually slow and the spreading diameter of the pastes maintain excellent stability, reflecting the positive influence of IBMA on the fluidity of NHL pastes. During continuous hydration process, the flocculation of NHL particles and the conversion of free water into encapsulated water are responsible for the reduction in the fluidity of the paste with time. Along with the encapsulated water becoming free in the system, the addition of IBMA to the NHL paste leads to the decomposition of the flocculated NHL structures [31]. It is suggested that IBMA may have a specific blocking impact on the early hydration of NHL, and the addition of IBMA allowed the NHL pastes to maintain fluidity for a longer period of time, which has a positive effect on its early fluidity.

### 3.3. Rheological Behavior

The rheological behavior of fresh NHL pastes is similar to that of other cementitious materials with shear thinning properties [28]. The rheological curves of fresh NHL pastes are shown in Figure 7, which indicate that the shear stress of the NHL pastes with different contents of IBMA decreases significantly along with the increase of IBMA content. Due to the NHL pastes exhibiting prominent Bingham characteristics after the addition of IBMA, the rheological behavior of the pastes was evaluated in the Bingham model by fitting the pastes’ rheological curve with two essential parameters, yield stress and plastic viscosity [21,32,33]. Therefore, each point in the rheological curve shown in Figure 7 was fitted according to the Bingham model, presented earlier in the text as Equation (1) in Section 2.3.4. The fitting curves of yield stress and plastic viscosity obtained are shown in Figure 7, from which it can be clearly observed that with the presence of IBMA (0–0.4%), both yield stress and plastic viscosity were diminishd. This indicates that yield stress and plastic viscosity are negatively correlated with flowability at low A content, similar to the results of some studies [34]. The addition of IBMA not only opened the flocculation structure, but also increased the Brownian motion of NHL particles in the pastes. The Brownian motion partly weakened the interaction between particles and kept the particles away from each other, which reduced the resistance to inter-particle motion and finally led to the reduction of yield stress and plastic viscosity. The yield stress even decreases to a value close to zero, which indicates that the NHL paste changes from being a plastic fluid to an approximately Newtonian fluid upon the addition of a sufficient content of IBMA. However, upon the further addition of IBMA, after the yield stress reached its lowest value, no further changes were observed, with the plastic viscosity then showing an increasing trend. One possible reason is the bridging effect of IBMA molecules between adjacent NHL particles, which increases the plastic viscosity but not enough to affect the yield strength. This change shows that in excess IBMA may act as a plasticizer, thus reinstating the original plasticity of the NHL pastes.

The rheological behavior of NHL pastes is also highly related to the volume fraction and dispersion of the system. According to the results of rheological tests, the addition of IBMA results in a decrease in the volume fraction of solid in the NHL pastes system and improves the particle dispersion in the system. At the same time, this leads to a reduction in the yield stress and plastic viscosity of the NHL pastes, which improves the rheological behavior of the pastes. Meanwhile, this indirectly demonstrates that the adsorption of IBMA on NHL particles changes the surface electrical properties of the particles.

### 3.4. Effect of IBMA on the Particle Dispersion of NHL Pastes

To determine the influence that IBMA has on the dispersibility of NHL particles, the microstructures of diluted NHL pastes were observed in the absence and presence of the IBMA copolymer. Fluidity measurements show that the optimal IBMA content in the NHL pastes is 0.6 wt%. Therefore, an NHL paste with 0.6 wt% IBMA content was selected as the control sample. Figure 8 shows the particle size distribution of the NHL paste suspension. By looking at the microstructure of the NHL paste, it can be noted from Figure 8c,d that in the paste without IBMA content, there are a large number of flocculated structures of different sizes with irregular edges formed by the adsorption of tiny particles around the large particles. The insides of the flocculated structures contain water. In the presence of IBMA, the flocculated particles gradually decompose and disperse, with some of the small particles not being adsorbed around the edges of the large particles and instead being relatively evenly distributed in the system. From the results, it is apparent that IBMA breaks down the flocculated structures, weakening the agglomeration of the NHL particles and improving their dispersion in the system. This phenomenon can be attributed to the effects of electrostatic repulsion and steric hindrance, which completely disperse the particles in the system and release the water trapped in the flocculated particles, increasing the volume of the free water phase and reducing the solid volume fraction [35]. These effects of IBMA make it difficult for flocculated structures to be generated, leading to a reduction in the yield stress and plastic viscosity of NHL. Moreover, these results are consistent with the previous results and together explain the changes in the rheological behavior and fluidity of NHL.

From analysis of the adsorption behavior results, it can be concluded that the dispersion of the NHL particles may be strongly related to the adsorption of IBMA on their surface. Through the –COO^–^ group, IBMA can adsorb on the NHL particles, which changes the electrical properties of the NHL surface and makes the particles repel each other due to electrostatic interactions, thus meaning that no further flocculated structures can form in the NHL pastes and those still existing structures cannot be broken down. Eventually, the water content in the flocculated structures is reduced, leading to an increase in free water content, and the even distribution of smaller particles throughout the system, which significantly affects the rheological properties of the paste. A microstructure model of fresh NHL pastes showing the effects of IBMA is shown in Figure IBMA has an effect on the microstructure of the fresh NHL paste, and an imitation microscopic mechanism is shown in Figure 9.

### 3.5. Hydration Heat Evolution Process of NHL Pastes

Since the hydration reaction is exothermic, monitoring the heat flux can lead to an understanding of the initial hydration of NHL. Thus, the influence that IBMA has on the heat given off by the hydration of NHL pastes containing different IBMA content was explored. The hydration evolution results are shown in Figure 10 and Figure 11. Along with the adsorption of IBMA on the surface of the NHL particles, the hydration kinetics curves of the NHL paste changed considerably, in accordance with the adsorption results mentioned previously. Figure 10 shows the total hydration exotherms of NHL samples with varying IBMA content at 20 °C. The cumulative hydration heat of the NHL sample with 0 wt% IBMA content over 168 h is 42.46 J/g, slightly different from the cumulative hydration heat measured in previous studies, but with both values being much lower than that of ordinary Portland cement [12,27]. This is due to the amount of C_3_S in NHL being negligible, with a small amount of C_3_A in the paste being hydrated and dissolved at the beginning of the reaction. The hydration of NHL over a 7-day period only involves the reaction of C_2_S and a small amount of undigested CaO. The total area of the hydration exotherm of the NHL paste gradually decreases with an increase in IBMA, proving that the incorporation of IBMA inhibits the entire hydration process. As can be seen from Figure 11, in the presence of IBMA, the hydration induction period gradually increases, and the hydration heat peak and exothermic rate declines, maybe because the adsorption of IBMA on the NHL particles prevents water and ions from making contact with NHL. Moreover, a new hydration peak gradually appears in the period of 0–10 h, which is very similar to the change in the second hydration exothermic peak. The reason for this may be because C_3_A and CaO are partially encapsulated, resulting in a delay in the release of heat. Surprisingly, the second exothermic peak of the NHL paste with 0.8 wt% IBMA was not detected due to the peak showing a delayed appearance after 7 days. Moreover, the chelating effect of Ca^2+^ and –COO^–^ inhibits the nucleation and growth of hydration products to some extent.

### 3.6. XRD, FT-IR Spectroscopy, and XPS Analysis

The evolution of hydration products upon the addition of different IBMA content to NHL was analyzed by conducting XRD and FT-IR spectroscopy measurements. The XRD patterns of NHL with different IBMA content are shown in Figure 12, from which it can be seen that the crystalline phases of the pure NHL sample cured for 7 days are mainly composed of calcium hydroxide (Ca(OH)_2_), calcite (CaCO_3_), quartz (SiO_2_), and unhydrated C_2_S. Since the low crystallinity of the hydrated product (C–S–H), it was not easy to determine the hydrated mineral structure in a mixed system by XRD, so the hydrated product was supplanted by the characteristic peak changes of Ca(OH)_2_ and C_2_S to analyze of the hydration process. It can be observed that the addition of IBMA did not produce any new phases in the NHL pastes. Compared with a pure NHL paste, the addition of IBMA makes the resulting NHL with IBMA content exhibit a gradual increase in a peak characteristic of C_2_S. In addition, it can be clearly seen that the intensity of the characteristic peak of Ca(OH)_2_ decreases. These changes indicate that the hydration process of C_2_S would be inhibit by the introduction of IBMA, thus reducing the formation of the hydration products C–S–H and Ca(OH)_2_ in the system. This results in an increment in the C_2_S production along with a reduction in the Ca(OH)_2_ production, thus slowing down the hydration process.

To further analyze the effect that IBMA has on the NHL hydration process, FT-IR spectroscopy was used to observe the changes in the NHL samples according to their different IBMA content. The FT-IR spectra of NHL samples with different IBMA content after 7 days of curing are shown in Figure 13. In the FT-IR spectra, the peak at 3634 cm^−1^ is caused by the stretching vibration of the hydroxyl group, which corresponds to the Ca(OH)_2_ in the sample. It is known from the literature research that the absorption peak adjacent to 999 cm^−1^ is the characteristic peak of C–S–H. Compared to the FT-IR spectrum of pure NHL paste with 0 wt% IBMA, this into the absorption peaks of Ca(OH)_2_ and C–S–H showed a decreasing trend with the increase of IBMA content, indicating that the content of Ca(OH)_2_ and C–S–H gradually decreases. This further proves that the retardation of hydration process in the NHL paste is caused by IBMA. The phenomenon above is that IBMA adsorbs on the hydration reaction mineral to form an adsorption layer, which to a certain extent hinders the mineral phase in NHL from making contact with free water. At the same time, an excess of IBMA is also adsorbed on the surfaces of the hydration products C–S–H and Ca(OH)_2_ via bridging with free Ca^2+^ ions in the aqueous solution, which also affects the diffusion of free water and ions at the interface between the solution and solid phase. Eventually, these effects lead to a delaying of the hydration process. In addition, the chelation of IBMA with free Ca^2+^ ions also leads to a decrease in the production of hydration Ca(OH)_2_ in the system, as the Ca^2+^ ions are a key component of the hydration kinetics [36,37].

XPS was employed to study the changes in the chemical compositions and electronic structures of pastes before and after the adsorption of IBMA, where the C 1 s XPS spectra before and after adsorption are shown in Figure 14. The 284.8 eV position corresponds to adventitious carbon, the peak area of which before and after adsorption remained unchanged. The peak at 286.5 eV can be attributed to O=C–OH or C=O bonds. The area under the peaks of the sample increased from 2212.763 before adsorption to 5903.991 after adsorption, indicating an increase in the O=C–OH or C=O bond content, due to IBMA being rich in O=C–OH or C=O bonds. The peak at 289.4 eV corresponds to C=O bonding. The area under the peaks of the sample increases from 26,875.7 before adsorption to 33,262.4 after adsorption, indicating an increase in the C=O bond content [38], suggesting that CaCO_3_ is exposed to form carbonyl groups. The peaks at 293.06 eV and 296.0 eV change from low intensity to high intensity and then back to low intensity. This phenomenon occurs because upon adding water to NHL, the CaCO_3_ is wrapped in NHL, so the C=O and –OH content is low, whereas after adding water, the CaCO_3_ is exposed, the area under the peaks of C=O and –OH increases, and CaCO_3_ is adsorbed, leading to an increase in peak intensity. Moreover, after the addition of IBMA, surface bonding occurs. The carbonyl C=O groups in IBMA and the hydroxyl –OH groups in the hydrate form hydrogen bonds. As an adsorption layer of IBMA is formed on NHL, the intensity of these peaks is thus reduced. The N 1 s XPS spectra before and after adsorption are shown in Figure 14. After adsorption, the peaks at 405.2 and 398.4 eV correspond to NO and NC, respectively, and it is evident that the area under the peaks is increased compared to that before adsorption. This phenomenon is due to the adsorption of IBMA by the NHL paste. Medium amino and imide. In short, it can be concluded from the XPS analysis that IBMA is adsorbed by the NHL paste and forms an adsorption layer on its surface.

The dispersion capability of IBMA was found to be highly correlated with its adsorption on the surface of the NHL particles. Meanwhile, research has shown that –COO^–^ groups can be adsorbed on the surface of mineral particles via electrostatic forces, or adsorbed on particles via the bonding together of –COO^–^ groups and Ca^2+^ ions [27]. Therefore, the adsorption behavior of IBMA can be strongly attributed to its molecular structure.

### 3.7. Mercury Intrusion Porosimetry (MIP)

MIP testing of NHL samples with different IBMA content was conducted. The changes in the pore size distributions and total porosities of NHL samples with different IBMA content after curing for 7 days are shown in Figure 15. Generally, capillary pores (≤1000 nm) originate from the space filled by unhydrated substances in the hardened paste, which was originally filled with water. Air pores (≥1000 nm) are usually pores introduced by the air entrained during the paste stirring process or due to additives such as surfactants. The same trend is also observed in this work, with an excessive addition of IBMA (≥1%) increasing the total porosity of the NHL paste (Figure 15). The pore size distribution NHL paste with no IBMA content shows that the capillary peak is observed in the range of 100–1000 nm, with no obvious stomatal peak. However, upon the addition of IBMA, the capillary pore peak gradually decreases and shifts to a smaller pore size. In addition, the presence of IBMA also leads to the appearance of pores (peaks above 1000 nm), and the associated peaks gradually grows in intensity. It can be seen from the above phenomenon that the addition of IBMA effectively improves the dispersibility of the NHL particles in the fresh paste, so that the space between the particles originally filled by water decreases, leading to a reduction in the size and number of pores in the system. In addition, the addition of IBMA also causes serious plasticization of the NHL paste, resulting in more air being introduced during stirring, which in turn leads to an increase in the number of pores in the system.

### 3.8. Mechanical Properties

The effect that IBMA has on the mechanical properties of NHL paste was studied. As shown in Figure 16, after being cured for different time periods, the compressive strength of the NHL sample first increases and then decreases with the addition of IBMA. The compressive strengths of the pure NHL sample are 1.80 MPa, 2.65 MPa, and 4.35 MPa after 7, 28, and 56 days, respectively. Upon the addition of IBMA, the compressive strength of NHL first increases and then decreases. When the IBMA content is 0.5 wt%, the compressive strengths of the samples are 2.5 MPa, 3.5 MPa, and 4.0 MPa after 7, 28, and 56 days, and the compressive strengths increased by 10%, 20%, and 10%, respectively, compared with those of the pure NHL sample. Therefore, the best mechanical properties of the NHL paste were obtained when the IBMA content was 0.5 wt%. So it was concluded that IBMA improves the compressive strength of NHL when the content of IBMA is in the range of 0.1–0.5 wt%, which may be attributed to the full growth of C–S–H crystals. Although IBMA inhibits the hydration process of C_2_S, it enables the C–S–H gel to fully crystallize and grow, and thus develop a dense network structure. Thus, the addition of IBMA not only improves the fluidity of the NHL paste but also increases its strength. Moreover, this also proves that IBMA is effective for improving the grouting properties of NHL.

## 4. Conclusions

In this study, the effects that IBMA have on the rheological properties and early hydration of NHL were explored. Moreover, the charge characteristics of IBMA and its adsorption behavior and dispersion mechanism in NHL pastes were revealed, with the following conclusions reached. IBMA exhibits multilayer (double-layer) adsorption on the surface of NHL. IBMA is adsorbed on NHL via electrostatic attraction, which results in a decrease in the zeta potential of the paste from +4.9 mV to a negative value. The reduction in the zeta potential is related to the adsorption of the first layer of IBMA, with the second layer of adsorption exhibiting no further changes on the zeta potential, which may be attributed to the complexation of –COO^–^ and Ca^2+^ in IBMA. IBMA significantly improves the fluidity of NHL paste, which reaches a stable state at saturation. The increase in the fluidity of NHL paste in the presence of IBMA can be positively correlated to the adsorption of the first layer of IBMA on the NHL, with the adsorption of a second layer of IBMA on NHL not making any further contribution toward enhancing the fluidity. The adsorption of IBMA on the NHL particles reduces the cross-linking of NHL particles, resulting in the decomposition of flocculated structures and greater dispersion of the NHL paste. This results in a reduction in stress and viscosity in the case of a low IBMA doping content, further improving the rheological properties of the NHL paste. The addition of IBMA delays the appearance of the hydration peak of the NHL paste and decreases the rate of hydration heat release, postponing the early hydration of the paste. However, the addition of a specific amount of IBMA significantly improves the early strength of NHL.

## Figures and Tables

**Figure 1 polymers-14-04104-f001:**
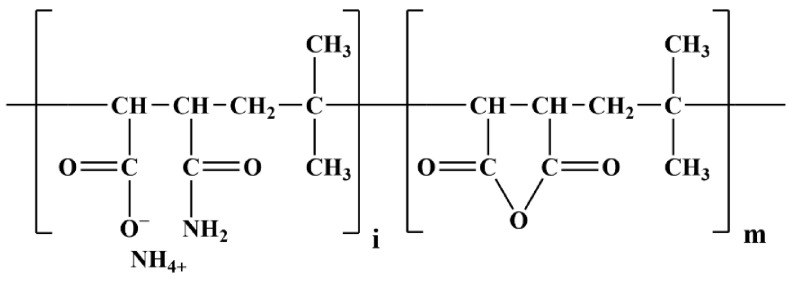
The chemical structure of IBMA.

**Figure 2 polymers-14-04104-f002:**
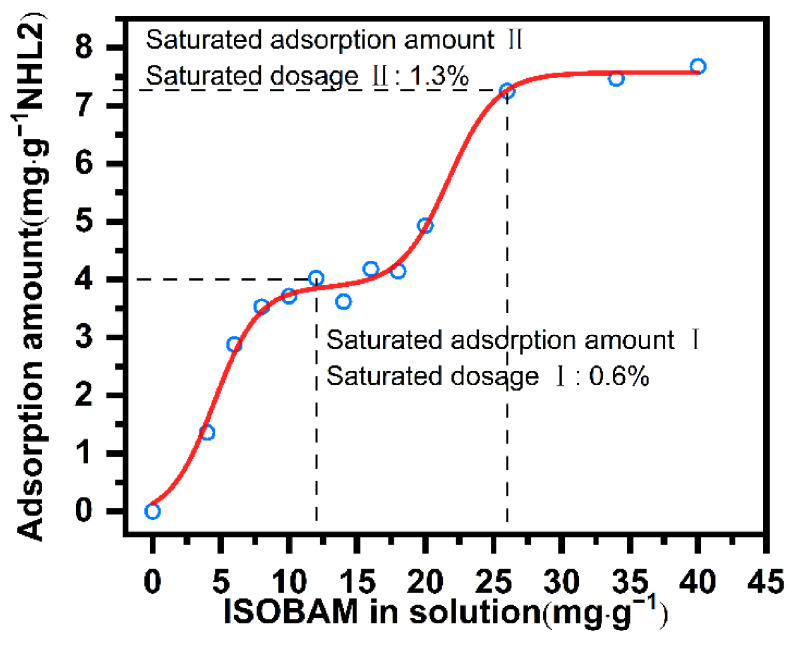
Isotherm of the adsorption of IBMA on the surface of NHL particles.

**Figure 3 polymers-14-04104-f003:**
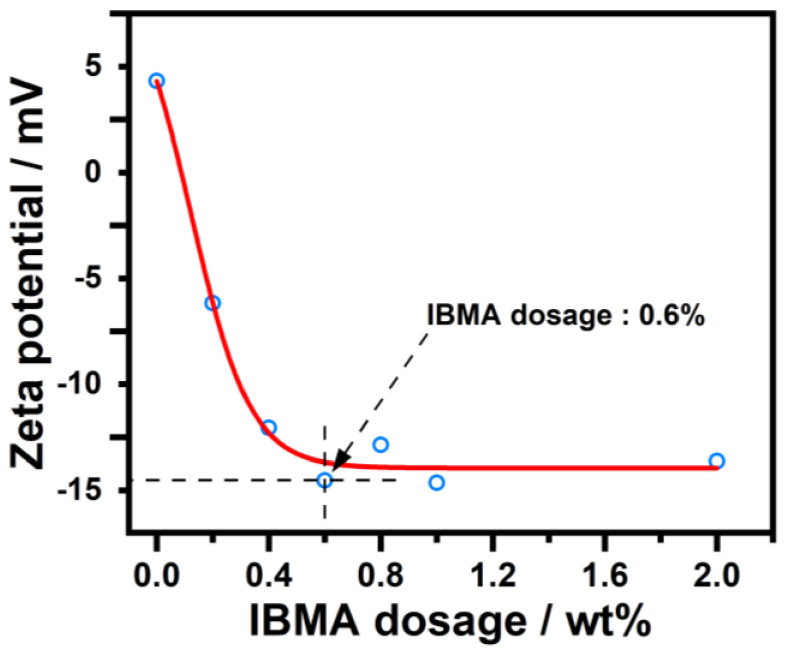
Zeta potential measurements of NHL pastes containing varying IBMA content.

**Figure 4 polymers-14-04104-f004:**
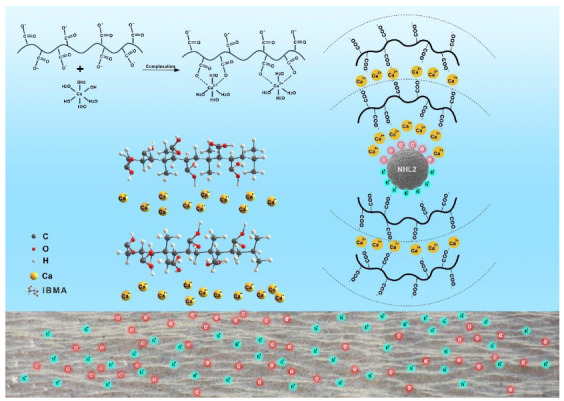
Schematic diagram of the structure of the chelate complex of IBMA.

**Figure 5 polymers-14-04104-f005:**
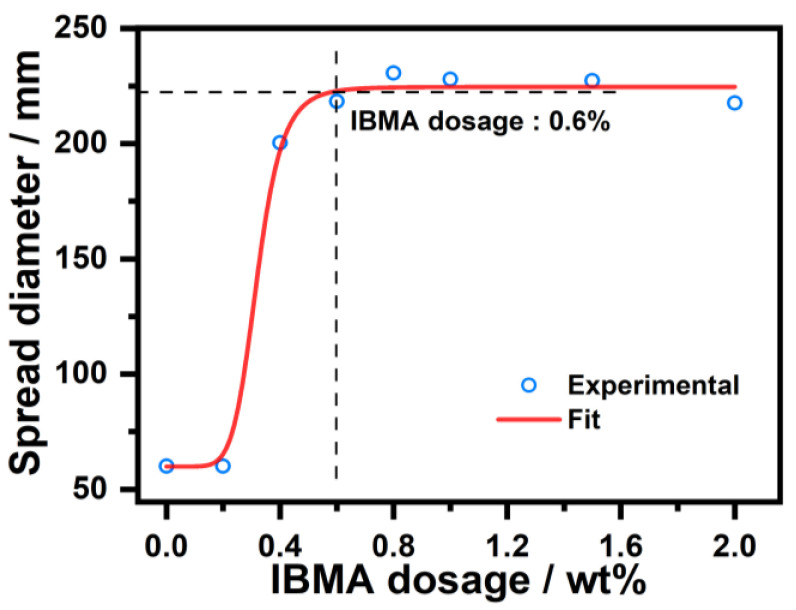
Spread diameter of fresh NHL pastes with varying IBMA content.

**Figure 6 polymers-14-04104-f006:**
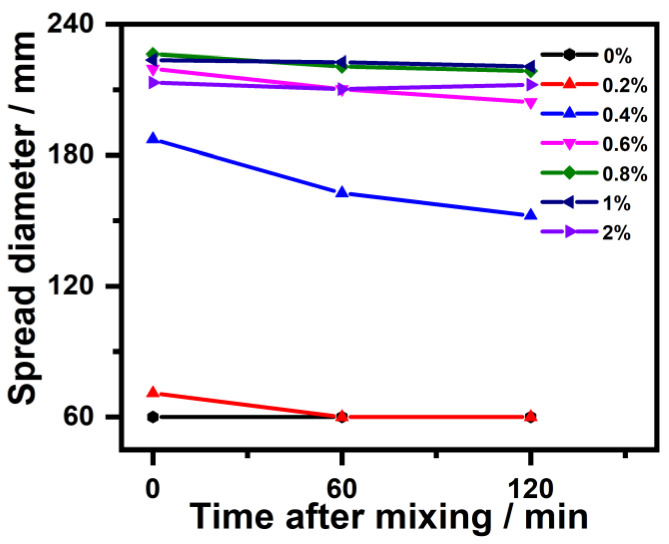
Changes in the spread diameter of NHL pastes with varying IBMA content with time.

**Figure 7 polymers-14-04104-f007:**
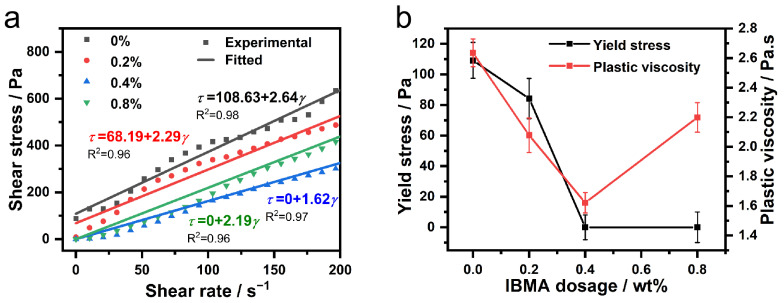
Rheological properties of fresh NHL pastes. (**a**) Rheological measurements and fitting curves of the IBMA–NHL pastes. (**b**) Yield stress and plastic viscosity of fresh NHL pastes with varying IBMA content.

**Figure 8 polymers-14-04104-f008:**
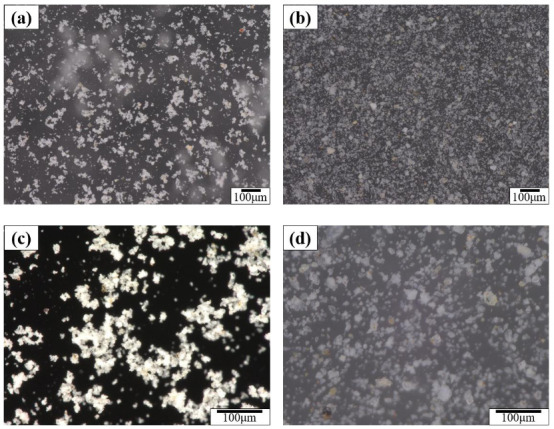
Optical microscopy images of NHL suspensions: (**a**,**c**) 0 wt% IBMA; (**b**,**d**) 0.6 wt% IBMA.

**Figure 9 polymers-14-04104-f009:**
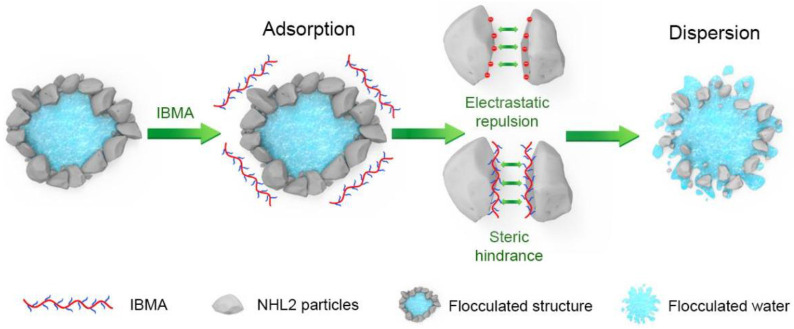
Schematic illustration of the effect that IBMA has on the microstructure of an NHL paste.

**Figure 10 polymers-14-04104-f010:**
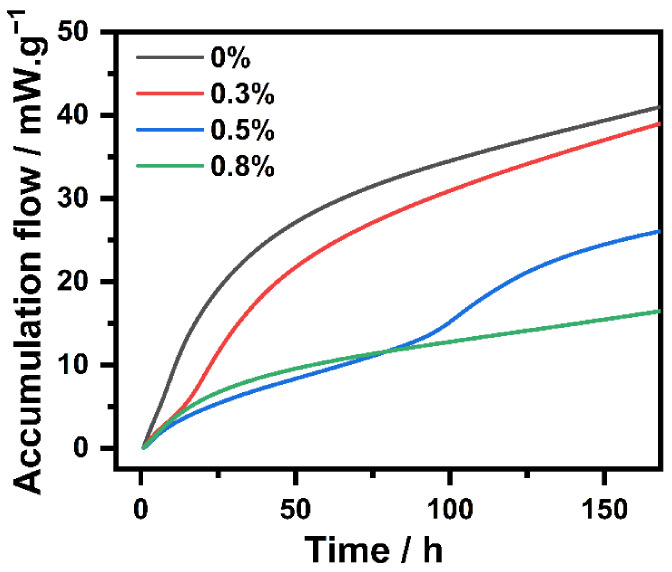
Influence that IBMA content has on the accumulation flow of the hydration of NHL.

**Figure 11 polymers-14-04104-f011:**
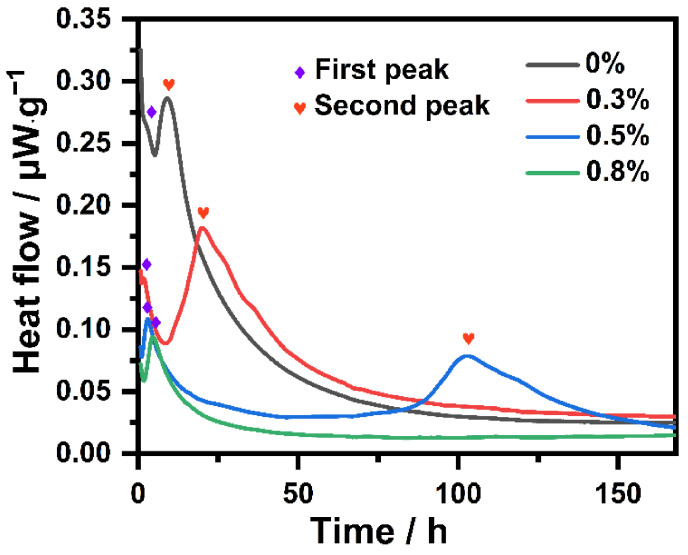
Influence of IBMA content on the hydration heat release rate of the hydration of NHL.

**Figure 12 polymers-14-04104-f012:**
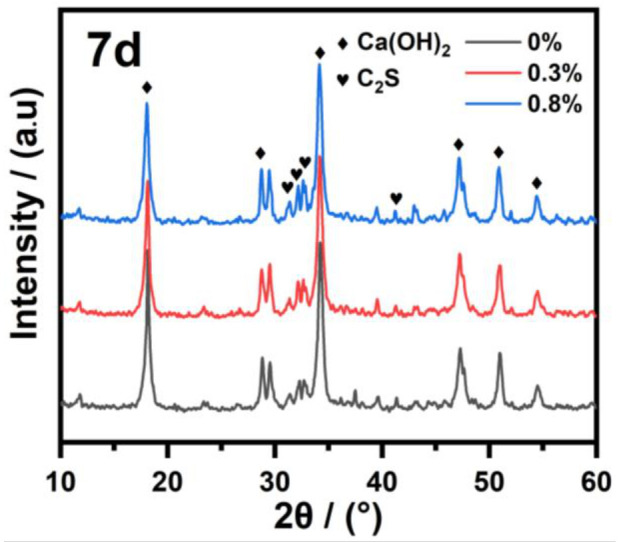
XRD patterns of NHL pastes with different IBMA content after 7 days of curing.

**Figure 13 polymers-14-04104-f013:**
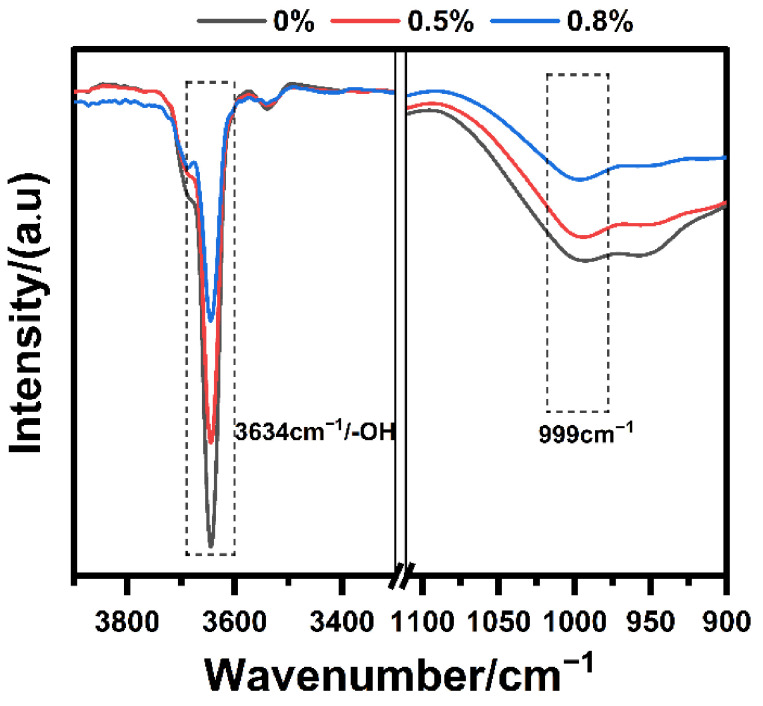
FT-IR spectra of NHL samples with different IBMA content after 7 days of curing.

**Figure 14 polymers-14-04104-f014:**
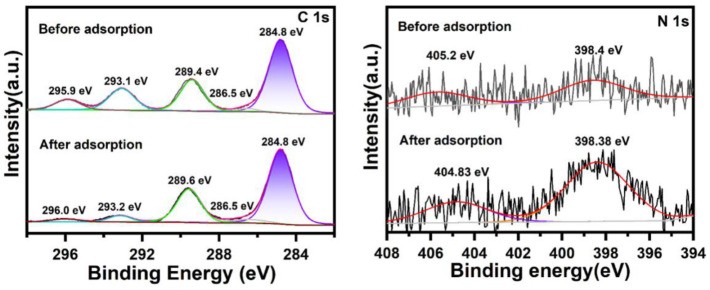
XPS patterns of NHL pastes with different IBMA content.

**Figure 15 polymers-14-04104-f015:**
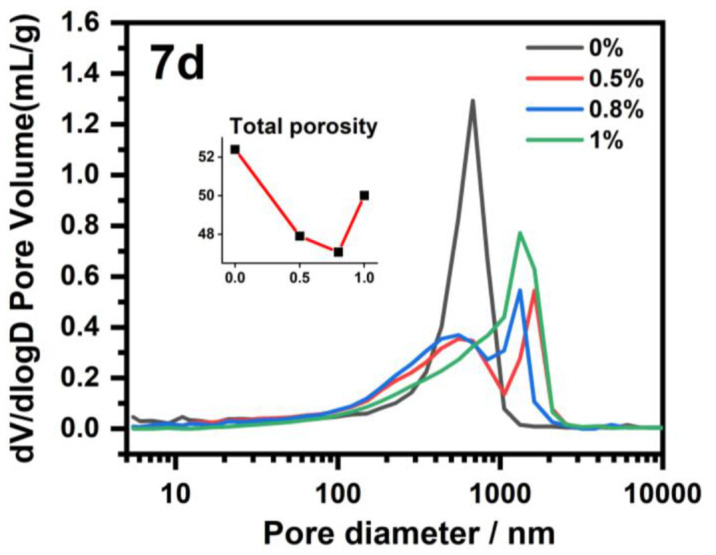
Pore size distribution measurement of NHL pastes with varying IBMA content by MIP testing.

**Figure 16 polymers-14-04104-f016:**
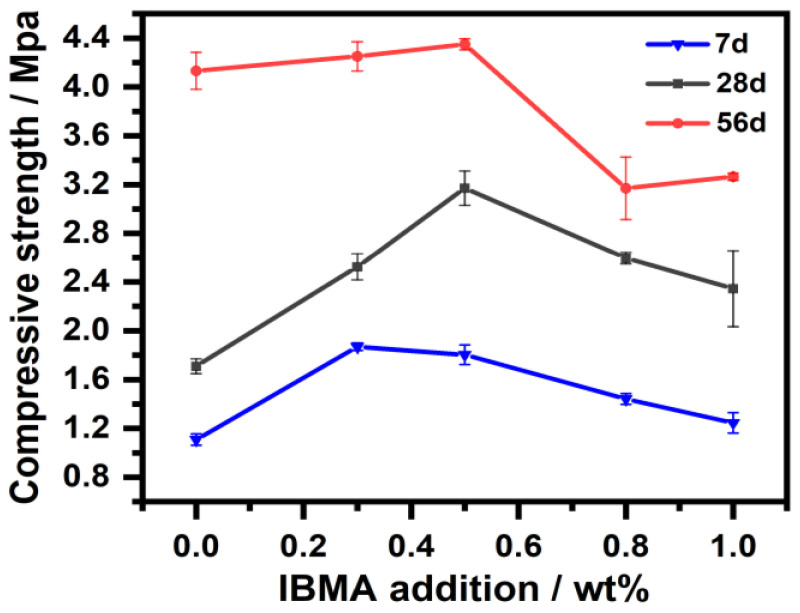
Compressive strength of NHL pastes with varying IBMA content after being cured for different time periods.

**Table 1 polymers-14-04104-t001:** Chemical composition of NHL2 (wt%).

**Elements**	CaO	SiO_2_	MgO	Al_2_O_3_	Fe_2_O_3_	SO_3_	K_2_O
**NHL2**	68.90	14.53	8.66	4.00	1.66	1.25	0.78

## Data Availability

All data generated or analyzed during this study are included in this published article.

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
