# Peer review of "Effects of an Isobutylene–Maleic Anhydride Copolymer on the Rheological Behavior and Early Hydration of Natural Hydraulic Lime"

_polymers, 2022, doi:10.3390/polym14194104_

Round 1
Reviewer 1 Report
This paper deals with the effects of an Isobutylene–Maleic Anhydride Copolymer addition on the Rheological Behavior and Early Hydration of Natural Hydraulic Lime matrix.
The work is interesting and there are several results presented.
1. The authors should use the appropriate terminology. "Fissures" is not used, cracks or cracking is more appropriate.
2. Mortar is referred to a mix including sand. However, sand is not used for the preparation of the samples. Therefore, the terminology "mortar" should be changed to "paste" throughout out the entire manuscript.
3. the numbers and the fonts at image 7 are very small and should be corrected. Also, at Fig. 7a a concentration of 0.8 is mentioned and at 7b this changes to 0.6.
4. The plastic viscosity value of the 0.4% sample shown in Fig. 7c it seems to be incorrect. According to the equations shown in fig. 7b, it should be 7.86.
5. The authors should explain why the samples with a content of 0.4% IBMA demonstrate lower shear stress compared to the ones having double amount (0.8%).
Reviewer 2 Report
The manuscript provides a characterisation of NHL materials when modified with an isobutylmaleic anhydride copolymer. Rheological behaviour and hydration are supposed to be the focus of the work.
Interestingly, the study shows an optimal composition for improving the fluidity of this system without negatively influencing the mechanical properties, also implemented in terms of the compressive strength.
The study is interesting, but, in my opinion, needs some revision in the case of the presentation and discussion of the rheological results. My suggestions are the following:
Regarding the way the rheological measurements were carried out:
-The named "round plate-to-plate die geometry", is it parallel-plate geometry or is it a special geometry? Has it been used a smooth or rough surface, to avoid the typical sliding of these materials?
The description of the procedure is a bit confusing, consider rewriting it for better understanding, in particular this sentence: After pre-shearing of and increasing the shear rate from 0 to 200 s-1 the shear rate was maintained for enough time to conduct steady-state measurements.
In the Bingham equation, the symbol for the shear rate parameter must be changed; γ ̇ (because γ is the strain).
There is an error in the units of plastic viscosity, it appears in the description of the Bingham equation and also in figures 7: the viscosity data in mPa s should be Pa s.
The titles in figure 7 should be corrected: shearing stress (pa) to shear stress (Pa), shearing rate (s-1) to shear rate (s-1).
In figure 7a, it can be seen that the different copolymer compositions in the NHL2 pastes not only changes the values of yield stress and plastic viscosity, but also the flow behaviour which is very different (as already explained in the text). In fact figures (a) and (b) should be shown as one figure, with symbols for the experimental data and lines for the model. It would be good to show the deviation from the Bingham fit and if necessary, as seems to be the case, to propose other flow models.
The discussion of the rheological data, in particular the paragraph on the relationship between yield stress - dispersion quality - reduction of the volume fraction of solids, should be rewritten: perhaps additional discussion should be included, not only based on previous results in the literature: It could be for example something about the change in the microstructure of the dispersion, actually observed in Fig. 8, or, if feasible, results of possible changes in thixotropic behaviour... ....
It should also be reviewed:
-Natural hydraulic lime is abbreviated as NHL2 at the beginning, but NHL2 is the commercial product labelled NHL2., isnt it? It is confusing.
Some error in the references: -Reference [20] should be : Luis G Baltazar et al. or Baltazar et al .; -Reference [21] : in the text: Fernando Jorne et al [21] , but in the list : [21] Aiad I. Influence of time addition of superplasticizers on the rheological properties of fresh cement pastes. Cem. Concr. Res. 2003, 33, 1229- 1234.

Round 2
Reviewer 1 Report
The manuscript can be published in the current form